# Studies of Nickel-Rich LiNi_0.85_Co_0.10_Mn_0.05_O_2_ Cathode Materials Doped with Molybdenum Ions for Lithium-Ion Batteries

**DOI:** 10.3390/ma14082070

**Published:** 2021-04-20

**Authors:** Francis Amalraj Susai, Daniela Kovacheva, Tatyana Kravchuk, Yaron Kauffmann, Sandipan Maiti, Arup Chakraborty, Sooraj Kunnikuruvan, Michael Talianker, Hadar Sclar, Yafit Fleger, Boris Markovsky, Doron Aurbach

**Affiliations:** 1Department of Chemistry, Institute for Nanotechnology and Advanced Materials (BINA), Bar-Ilan University, Ramat-Gan 52900, Israel; sfamalraj@gmail.com (F.A.S.); maiti.sandipan@biu.ac.il (S.M.); arupchakraborty719@gmail.com (A.C.); soorajscs@gmail.com (S.K.); Hadar.Sclar@biu.ac.il (H.S.); Yafit.Fleger@biu.ac.il (Y.F.); 2Institute of General and Inorganic Chemistry, Bulgarian Academy of Sciences, 1113 Sofia, Bulgaria; dkovacheva@gmail.com; 3Solid State Institute, Technion—Israel Institute of Technology, Haifa 32000, Israel; ktatyana@si.technion.ac.il; 4Department of Materials Science and Engineering, Technion—Israel Institute of Technology, Haifa 32000, Israel; mtyaron@tx.technion.ac.il; 5Department of Materials Engineering, Ben-Gurion University of the Negev, Beer-Sheva 84105, Israel; mtalianker973@gmail.com

**Keywords:** lithium-ion batteries, LiNi_0.85_Co_0.10_Mn_0.05_O_2_ cathode materials, Mo-doping, cycling behavior, dopant segregation at the surface

## Abstract

In this work, we continued our systematic investigations on synthesis, structural studies, and electrochemical behavior of Ni-rich materials Li[Ni_x_Co_y_Mn_z_]O_2_ (x + y + z = 1; x ≥ 0.8) for advanced lithium-ion batteries (LIBs). We focused, herein, on LiNi_0.85_Co_0.10_Mn_0.05_O_2_ (NCM85) and demonstrated that doping this material with high-charge cation Mo^6+^ (1 at. %, by a minor nickel substitution) results in substantially stable cycling performance, increased rate capability, lowering of the voltage hysteresis, and impedance in Li-cells with EC-EMC/LiPF_6_ solutions. Incorporation of Mo-dopant into the NCM85 structure was carried out by in-situ approach, upon the synthesis using ammonium molybdate as the precursor. From X-ray diffraction studies and based on our previous investigation of Mo-doped NCM523 and Ni-rich NCM811 materials, it was revealed that Mo^6+^ preferably substitutes Ni residing either in *3a* or *3b* sites. We correlated the improved behavior of the doped NCM85 electrode materials in Li-cells with a partial Mo segregation at the surface and at the grain boundaries, a tendency established previously in our lab for the other members of the Li[Ni_x_Co_y_Mn_z_]O_2_ family.

## 1. Introduction

Recently, electric vehicles have already become a reality around the world, so development of power sources, which guarantee their long driving ranges and safety is critically important. Among these power sources, advanced lithium ion batteries (LIBs) are the most appropriate ones for electrochemical propulsion, since they may provide the necessary high energy density, cycle life, stability, and reasonable safety [1,2,3,4,5,6,7,8,9,10,11]. The limiting factor of energy content in LIBs are the positive electrodes (cathodes) and their improvement is much more effective compared to other parts of batteries to get high energy density [12]. 

In this paper, we focused on cathodes comprising layered structure lithiated oxides (the space group of *R-3m*) of the general formulae Li[Ni_x_Co_y_Mn_z_]O_2_ (x + y + z = 1) with high Ni-content of ≥80 at. %. These materials (NCM) have attracted attention since they exhibit specific capacity ≥ 200 mAhg^−1^ due to the fact that nickel is the main electroactive species in the host structure (Ni^2+^ ↔ Ni^4+^) [4,5,6,13,14]. Moreover, they are considered as the most promising cathodes in advanced LIBs to enable longer driving-range electric cars [7,8]. However, several drawbacks of NCMs are low electronic and ionic conductivities, low structural stability, fast electrode capacity decay, high voltage hysteresis, and insufficient rate capability [4]. In addition, as the content of nickel and the specific capacity are higher, their structural stability is lower and capacity fading during cycling increases substantially [15,16,17,18]. That is why, significant efforts have been devoted in the field to improving intrinsic characteristics of NCM materials by lattice doping with mono- and multi-valence cations, like Ag^+^, Mg^2+^, Cu^2+^_,_ Al^3+^, Cr^3+^, B^3+^, Fe^3+^, Ti^4+^, Zr^4+^, Ta^5+^, W^6+^, and Mo^6+^ [19]. It is well accepted that doping can reduce Ni^2+^ ion migration into the Li-layer, preventing Ni^2+^/Li^+^ mixing during synthesis and electrochemical cycling, thus significantly stabilizing capacity behavior, lowering electrode impedance, and enhancing power performance as was shown for many NCM cathodes [3,20]. It was shown recently [18] that doping LiNi_0.85_Co_0.10_Mn_0.05_O_2_ (NCM85) with Aluminum (2 mol %) by partial substitution of Mn results in mitigation of this material’s structural degradation and capacity fade during cycling up to 4.8 V. We have also demonstrated that Al-doping (0.01 at. %) in NCM523 at the expense of Ni, Co, and Mn resulted in substantial improvements in the electrodes’ performance upon cycling and aging at 60 °C [21]. We further established that doping Ni-rich LiNi_0.6_Co_0.2_Mn_0.2_O_2_ material by Zr^4+^ cations resulted in higher stability, higher rate capability and lower charge-transfer resistance of doped electrodes [22]. 

Our DFT calculations have shown that Zr-doping partially inhibits layered-to-spinel structural transformation during cycling. This effect was ascribed to destabilization of Ni tetrahedral sites and reducing the number of Jahn–Teller active Ni^3+^ ions [22]. Although high-charge state dopants like Nb^5+^, Ta^5+^, W^6+^, and Mo^6+^ are promising for NCM materials, their effect has been less studied [23,24,25,26,27,28]. For instance, it was demonstrated by Kim et al. that stable performance can be achieved for cathodes, comprising Ni-rich (Ni ≥ 80 at. %) materials doped with minor level (1 mol %) of W^6+^ [29]. 

In a recent work by the Y.-K. Sun’s group [30], the authors compared the newly synthesized W-doped Li[Ni_0.9_Co_0.09_W_0.01_]O_2_ sample (NCW90) to the well-characterized Li[Ni_0.885_Co_0.1_Al_0.015_]O_2_ in which Al was substituted by W, and demonstrated its superior structural and thermal stability compared to the commercialized NCA cathode. NCW90 electrodes can deliver both high energy density and a long battery life attributed to the significantly modified cathode microstructure through particle refinement. It was shown in the literature that molybdenum substitution in Ni-rich materials (Ni = 80 at. %) suppresses the crystal structure transformation from spinel to rock salt, decreases the charge-transfer resistance, improves the capacity retention upon prolonged cycling, as well as the thermal stability of the doped samples in reactions with electrolyte solutions [5,24,31]. Moreover, Mo-modified Ni_0.815_Co_0.15_Al_0.035_O_2_ (NCA) Ni-rich cathodes demonstrated superior rate capability and cycling stability at 27 °C and 55 °C [32]. In a recent study, we highlighted that doping of the NCM811 cathodes with Ta^5+^ increased their long-term cycling stability at 45 °C, decreased interfacial resistance, and lowered the voltage hysteresis compared to the undoped material [27]. Importantly that Ni-rich NCM811 and Al-doped LiNi_0.8_Co_0.2−x_Al_x_O_2_ (NCA, 0 < x < 0.1) cathodes delivering up to 200 mAh/g at a slow rate can be considered now as reference electrodes for advanced LIBs. Though increasing the Ni-content to 85–90 at. % results in capacity up to 210–220 mAh/g [4], this will lead to increased structural instability as well as to pronounced capacity and voltage fade upon prolonged cycling. Therefore, stabilization of the above characteristics of Ni-rich cathodes is highly important and deserves further intensive investigations. 

Being motivated to study NCM materials with high-Ni content (>80 at. %), we synthesized LiNi_0.85_Co_0.10_Mn_0.05_O_2_ (NCM85) undoped and LiNi_0.84_Mo_0.01_Co_0.10_Mn_0.05_O_2_ doped with Mo^6+^ cations (1 at. %) by a simple, fast, and low cost solution-combustion reaction (SCR) [5]. Though this method produces submicronic NCM particles with pyramidal morphology different from that of micron-sized ball-shaped ones produced by other techniques and with capacities slightly lower compared to those obtained for instance, by co-precipitation synthesis [33,34], solution-combustion followed by annealing the product under pure oxygen at 740–760 °C can be successfully used for a fast screening in-situ cationic doping. Taking into account our previous methodical work on Mo-doped NCM523 and Mo-doped NCM811 [5,35] the dopant level in the newly synthesized Ni-rich NCM85 was also set up to 1 at. % to demonstrate positive impacts of this minor level dopant on the cathode cycling behavior, the voltage hysteresis, rate capability, impedance, and thermal stability in reactions with battery solutions containing EC, EMC, and LiPF_6_. It is important to emphasize that NCM materials with various Ni-content (50, 80, and 85 at. %) were produced in our lab using the same synthetic method, annealing protocol, and the same amount of the dopant (Mo^6+^) incorporated into the structure by an in-situ approach. The effect of the dopant segregation at the surface is also a common observation for these NCM materials. Therefore, the results, thus obtained by systematic studies, which demonstrated considerable advantages of the electrodes comprising Mo-doped samples are reasonably promising for this family of Li[Ni_x_Co_y_Mn_z_]O_2_ materials. Consequently, the present work, as well as our ongoing intensive studies of NCM with Ni ≥ 90 at. %, are undoubtedly important in developing optimized cathode materials for advanced LIBs.

This paper is organized as follows. In Section 2, we elaborated on the synthesized materials and synthesis protocol, preparation of electrodes, details of electrochemical measurements, as well as of the spectroscopic studies, and thermal stability tests of NCM85 materials in battery solutions. In Section 3, we discussed our results starting from structural examination followed by analysis of the electrochemical properties of NCM85 electrodes in Li-cells, thermal behavior and Mo-dopant segregation at the surface. Finally, we presented conclusions in Section 4.

It is important to emphasize that the goal of the studies described herein was not a presentation of a fully practical high specific capacity cathode material for advanced Li-ion batteries, but rather we examined herein a proof of concept.

We intended herein to demonstrate that adding 5% of Ni to Ni-rich NCM cathode materials beyond the Ni content of standard material like NCM811 or NCA (80% nickel), even when using a very simple synthetic mode, for instance, self-combustion reaction, provides the expected jump in specific capacity: >205 mAh/g compared to around 190 mAh/g for NCM cathode material with 80% Ni.

Moreover, it was important to realize and exhibit the pronounced stabilization effect of doping by a small amount of molybdenum on this new NCM85 cathode material. From preliminary studies, we observed that a Mo-doping level, as small as 1 at. %, can be considered as optimal in NCM85 material to enhance the electrochemical cycling performance, to lower electrode impedance and heat evolution in reactions with battery solutions, and stabilize reversible structural transformations. While a conclusive stabilization effect of Mo-doping was demonstrated with NCM523 and NCM811 cathode materials, it was important to show that even when we increased the concentration of nickel in the cathode material to 85% and used a relatively simple synthesis, the stabilization effect of doping with Mo was very clear and impressive.

## 2. Materials and Methods

### 2.1. Synthesis of Materials and Chemical Analysis

NCM materials with the following compositions LiNi_0.__85_Co_0.__05_Mn_0.__05_O_2_ (undoped) and Molybdenum doped LiNi_0.84_Mo_0.01_Co_0.10_Mn_0.05_O_2_ in the cost of Ni (1 at. %) were synthesized by solution combustion method as described previously [5,22,36,37]. We used the following precursors (analytical grade) for this synthesis: LiOH∙H_2_O, Ni(NO_3_)_2_∙6H_2_O, Co(NO_3_)_2_∙6H_2_O, Mn(NO_3_)_2_∙4H_2_O, H_24_Mo_7_N_6_O_24_∙4H_2_O (doping agent) and sucrose C_12_H_22_O_11_. A small excess of lithium was provided (Li1.02) according to our previous experience. The mechanism of self-combustion reactions to form lithiated transition metal oxides was already discussed in details [37,38]. This method is very convenient for fast combinatorial studies of many compositions with the desirable stoichiometry in lab scales [5,22,35]. We annealed NCM85 undoped and Mo-doped samples in a tubular furnace, at 760 °C in pure oxygen for 6 h. Chemical analysis of the synthesized materials was performed by the inductive coupled plasma technique (ICP-AES, spectrometer Ultima-2 from Jobin Yvon Horiba, Kyoto, Japan) and the results of the chemical analysis are presented in Table 1.

### 2.2. X-ray Diffraction (XRD) Measurements

The NCM85 undoped and Mo-doped samples were characterized by powder X-Ray diffraction (PXRD) using Bruker D8 Advance diffractometer (Bruker D8 Advance, manufacturer, Bruker AXS GmbH, Karlsruhe, Germany) with Cu Kα radiation and a LynxEye detector. PXRD patterns were collected within the range from 10 to 80° 2θ with a constant step 0.02° 2θ. Phase identification was performed with the Diffrac*^plus^* EVA (version 2, Bruker AXS GmbH, Karlsruhe, Germany) [39] using ICDD-PDF2 Database. Rietveld refinement procedures were performed with the Topas-4.2 software package (version is 4.1 Bruker AXS GmbH, Karlsruhe, Germany) [40]. The Rietveld refinement of the crystal structure includes the unit cell parameters, fractional atomic positions, isotropic thermal displacement parameters and occupancies of all atoms in the structure. Mean coherent domain size (crystallite size) of phases was obtained by analysis of the diffraction lines broadening. For this purpose, the profiles of the diffraction peaks were fitted by means of fundamental parameters approach implemented in the Topas-4.2.

### 2.3. Preparation of Electrodes and Electrochemical Cells

Working electrodes (cathodes) for electrochemical testing were prepared by casting slurry made from a mixture of the active material LiNi_0.__85_Co_0.__05_Mn_0.__05_O_2_ undoped or Mo-doped with carbon black super P, graphite KS-6 and polyvinylidene difluoride binder PVdF in a ratio of 88:4:4:4 wt.%, in N-methyl pyrrolidone, as described previously [5,35]. We used two-electrode cells of 2325 coin-type configuration and three-electrode home-made pouch-type cells (with Li-reference electrodes) for impedance measurements [41]. The geometric area of the working electrodes was ≈1.5 cm^2^ and ≈10.5 cm^2^, respectively, for these cells, and the average loading of the active electrode mass was 4–5 mg/cm^2^ corresponding to ≈1 mAh/cm^2^. The cells were filled with the electrolyte solution LP57 (Li battery grade, from BASF, Ludwigshafen, Germany) comprising 1 M LiPF_6_ dissolved in a mixture of ethyl-methyl carbonate and ethylene carbonate, EMC: EC (7:3 by weight).

### 2.4. Electrochemical Measurements

For statistical purposes, we studied electrochemical performance of at least 2–3 cells simultaneously of undoped and Mo-doped samples and the results were averaged. The accuracy of these measurements was ≈95%. All cells were cycled in the potential range of 2.8–4.3 V and subjected first to a formation procedure consisting of two charge–discharge cycles at a C/15 rate (constant current–constant voltage mode, CC–CV) providing potentiostatic step for 3 h at 4.3 V, at 30 °C (1C = 180 mAh/g). They were continuously cycled for measuring the rate capability at various current densities, at 30 °C followed by cycling at C/3 rate using CC–CV mode with a potentiostatic step for 30 min at 4.3 V. The equipment used was a multichannel Arbin battery cycler and a battery test unit model 1470, coupled with a FRA model 1255 [5,35]. The alternating voltage amplitude in impedance measurements was 3 mV and the frequency ranged from 100 kHz to 5 mHz. All the potentials in this paper are given vs. Li^+^/Li.

### 2.5. Differential Scanning Calorimetry (DSC) Measurements

The DSC measurements of NCM85 pristine powders were carried out at a heating rate of 1 °C∙min^−1^ in the temperature range of 30–320 °C using the Mettler Toledo Inc., Columbus, OH, USA, model DSC 3+ device. These tests were performed with reusable high-pressure gold-plated stainless-steel crucibles (diameter: 6 mm, volume: 30 µL). Typically, ≈3.5 mg of the undoped or Mo-doped cathode materials with ≈3 µL of LP57 electrolyte solution were loaded into the crucibles. The initial and final weights of the loaded crucibles were measured using an analytical balance from Mettler Toledo AB135-S/FACT in order to ensure no leaking of the materials has occurred during the experiments. Each DSC experiment was carried out 3 times.

### 2.6. Time-of-Flight Secondary-Ion-Mass-Spectroscopy (TOF-SIMS) Studies

These studies were performed using ION-TOF GmbH TOF.SIMS 5 (ION-TOF Gmbh, Munster, Germany). The depth profiles were taken in a dual mode using 25 KeV Bi^+^ analysis ions and Cs^+^ (for negative secondary’s)/O_2_^+^ (for positive secondary’s) as the sputtering ions (incident at 45°). The sputter rate was 0.15 nm/s. The powder samples of NCM-85 undoped and Mo-doped were deposited from ethanol suspensions onto round-shaped (1.3-cm diameter) gold plates. The sputtered area for all measurements was 500 × 500 μm^2^ and the acquisition area was 100 × 100 μm^2^.

### 2.7. Scanning Transmission Electron Microscopy (STEM) Measurements

Possible molybdenum segregation at the surface was studied using the cross-sections of the Mo-doped NCM85 material. They were prepared by a focus ion beam (FIB) using a Helios 600 dual beam instrument (Thermo Fisher, MA, USA). These samples were then investigated in double Cs-corrected HR-S/TEM Titan Themis G2 60–300 microscope (FEI/Thermo Fisher) operated at 300 keV. High-angle annular dark-field (HAADF) imaging combined with energy dispersive spectroscopy (EDS system with DualX detector, Bruker Corporation, MA, USA) were used to analyze the chemical composition and distribution of the various elements over the sample.

## 3. Results and Discussion

First, chemical analysis of NCM85 undoped and Mo-doped materials shows that their elemental compositions are very close to the desired ones with the dopant concentration of ≈1 at. % in LiNi_0.84_Mo_0.01_Co_0.10_Mn_0.05_O_2_ (Table 1). As it follows from XPS studies of Mo-doped sample, the typical Mo^6+^ spectrum of LiNi_0.84_Mo_0.01_Co_0.10_Mn_0.05_O_2_ sample (not shown here) displays characteristic Mo 3d_3/2_ and Mo 3d_5/2_ components at 235.5 and 232.1 eV, respectively [5].

Examination of the powder XRD patterns of these materials (Figure 1) demonstrates that they both are single phase possessing rhombohedral structure of R-*3m* symmetry group.

The lattice *a-* and *c*-parameters, mean coherent domain size, cationic mixing (Ni^2+^ in the Li-layer), and the calculated intensity ratios of 003, 104, and 012, and the 006 and 101 planes of these samples are presented in Table 2.

It is important to note that results of a minor Mo-doping have a slight increase of the lattice *a* and *c*-parameters and the same value of the *c/a* ratio. The increase of the unit cell parameters is ascribed to the compensative increase of the amount of Ni^2+^ in the doped sample in order to keep the electro-neutrality of the material. This fact is an indication for Mo incorporation into the structure. Having in mind the synthesis conditions of NCM85 (760 °C in pure oxygen for 6 h), the oxidation state of molybdenum is assumed to be 6^+^. In addition, we have shown in our previous combined experimental and computational modelling studies [5,35] that Mo ions replaced Ni ions out of other transition metal ions in Mo-doped NCM811 and NCM523 with the same dopant concentration of 1 at. %. From Density Functional Theory calculations, we concluded that the Mo ion was in the 6^+^-oxidation state and this highly charged cation changed the distribution of Ni-ions having different oxidation states and effectively Ni^3+^ ions were reduced upon Mo-doping. Therefore, we believe the same would be true in the case of 1 at. % Mo-doped NCM85 studied in the current work. As it follows from Table 2, the preserved degree of the cation mixing, clear splitting of the (108)/(110) diffraction peaks, and the lower value of (I_012_ + I_006_)/I_101_ ratio imply, thus, that the highly ordered layered structure of NCM85 was not only preserved but improved upon Mo doping. This may relate to the low crystallographic mismatch between ionic radii (r) of Ni-host (rNi^2+^ = 69 pm) and Mo-guest (rMo^6+^ = 59 pm). Table 2 also shows a decrease of the mean coherent domain size for the Mo-doped material. The smaller mean coherent domain size is an indication for the increased number of intra-granular defects (stacking faults, grain boundaries, twinning, antiphase boundaries, etc.) in doped sample compared to the undoped one. These defects may have positive effect on the lithium transport properties.

In Figure 2, we present the schematic for the unit cells of undoped and Mo-doped NCM85. Here, *3a* sites are occupied by transition metals (TMs) and *3b* sites by Li-ions in alternating layers along the *c*-axis based on our previous experimental and computational studies on other Ni-rich materials such as NCM523 and NCM811. Based on our earlier studies on Mo-doping in NCM811, which is quite close to NCM85 in its stoichiometry, Mo-dopant is expected to preferably substitute Ni-sites in the unit cell. As it was shown in our recent work [12], the amount of Ni^3+^ ions increases with increases of the Nickel concentration in NCM materials compared to Ni^2+^, and this makes the system unstable by increasing the number of Jahn–Teller active centers. Doping with high-valence state ion, like Mo^6+^ helps to improve the structural stability by reducing Ni^3+^ in NCM to maintain charge neutrality of the system [5,35]. We also demonstrated from analysis on electronic structure of Mo-doped NCM materials that Ni-*d* states are hybridized with O-*p* states and reside near the Fermi level indicating, thus, that Ni-ions are the more redox active ones [5,35]. Furthermore, it was noted that in Ni-rich NCM811 extra Mo-*d* states appear near the conduction band minima implying, thus, increasing conductivity. We expect the same effect for Mo-doped NCM85 and discuss this issue below.

We have established that in LiNi_0.84_Mo_0.01_Co_0.10_Mn_0.05_O_2_, the Mo-dopant tends to segregate at the outermost surface layer of ≈15−20 Å as demonstrated in Figure 3a that displays the depth profiles of Molybdenum obtained from ToF-SIMS studies. This finding is similar to that of LiNi_0.__50_Co_0.__20_Mn_0.__30_O_2_ and LiNi_0.__80_Co_0.__10_Mn_0.__10_O_2_ cathode materials doped with 1 at. % Mo studied in our recent papers [5,35]. The outermost surface layer in these NCM is enriched with the dopant, while its concentration decreases and levels off in the “bulk”. Note that due to high roughness of the powder NCM85 samples (though they were deposited and pressed onto flat gold plates), “surface” and “bulk” regions are shown only schematically, as an eye guide. In Figure 3b,c we present the results of our HAADF-STEM-EDS studies of the NCM85 Mo-doped material. These results clearly demonstrate that the dopant is homogeneously distributed inside the bulk of the grains while its concentration increases sharply at the grain boundaries (GBs) of NCM85 Mo-doped samples. Therefore, these findings undoubtedly indicate that the dopant preferential segregation occurs at the GBs of LiNi_0.84_Mo_0.01_Co_0.10_Mn_0.05_O_2_ doped with Mo^6+^ cations. According to the literature reports [42], the enrichment (segregation) of the dopant at GBs substantially affects the local structure and chemical and other macroscopic properties of the materials, therefore studies and controlling this phenomenon offer a promising approach for materials engineering. We assumed that the above-established phenomena of the Mo-segregation play a significant role in the electrochemical behavior of NCM85, due to the modified electrode/solution interface [5] and to the grain boundary effects on ionic transport as suggested in several reports [43,44,45]. The authors of these papers proposed two competing pathways of ionic conduction, for instance of mobile Li^+^ ions, in solids: the “granular” pathway representing conduction of these ions through the grains and GBs, and the “grain boundary” pathway that dominates when GB conduction is comparable to that in the bulk structure.

We present schematics of possible pathways of Li^+^-ions in NCM85 undoped and doped materials in Figure 4a,b, respectively, and propose that in the case of the undoped sample, the GBs are more resistive compared to the bulk, whereas for the Mo-doped sample the GBs conduction increases due to the dopant segregation. In this case, one may expect an improved performance and increased electrochemical kinetics of Li^+^-ions for electrodes comprising doped NCM85 material compared to those of undoped ones. We now discuss the comparative electrochemical performances of these electrodes in Li-cells, in terms of capacity retention, the voltage hysteresis, rate capability, and impedance. First, it was established that upon initial cycling at a C/10 rate, both undoped and Mo-doped samples exhibited typical voltage vs. capacity profiles [5,33], with charge capacities of 235 mAh/g for both electrodes (profiles are not shown here). The discharge capacities were calculated to be 206 and 216 mAh/g, respectively for undoped and doped samples and the corresponding irreversible capacity losses (ICL) of these electrodes were 12.5% and 8.2%. The lower ICL (the higher Coulombic efficiency) of the Mo-doped electrodes is in agreement with literature reports on the Mo^6+^ doped [5] and Nb^5+^ doped [28] NCM811 materials suggesting lesser side reactions of these electrodes due to the modified electrode/solution interface. It is evident from our DSC studies that in fact, doped electrodes exhibit lower by ≈25% the total heat Q_t_ = 470 ± 5 J/g evolved in thermal reactions of this NCM85 material with EMC: EC (7:3 by weight)/LiPF_6_ solution compared to Q_t_ = 630 ± 9 J/g measured for the undoped samples, as seen in Figure 5. Note that this thermo-chemical behavior is another common characteristic of Mo-doped NCM811 and NCM85 materials, established in our work, for their reactions with the battery solutions [5]. It can be suggested that higher heat evolved at ≈75 °C for NCM85-Mo doped samples (Figure 5b, zone I) relates, likely to the presence of Mo-containing species, for instance Li_2_MoO_4_, formed during the synthesis. This surface species can facilitate electrochemical performance of NCM electrodes, as demonstrated in the literature [46,47]**.**

Further, we demonstrated that Mo-doped LiNi_0.84_Mo_0.01_Co_0.10_Mn_0.05_O_2_ indeed exhibited an improved electrochemical behavior, namely more stable cycling with ≈15% higher capacity retention, much lower (by ≈60%) voltage hysteresis and its evolution, and higher rate capability at various C-rates, as presented in Figure 6a–c, Figure 7 and Figure 8, respectively. For instance, the rate capability of doped electrodes at a 2C rate exceeds that of the undoped by ≈13%, as follows from Figure 8. Faster electrochemical kinetics of Mo-doped electrodes upon cycling is evidenced also from the analysis of differential capacity plots (Figure 9 and Table 3). They clearly demonstrate that peak potentials of both anodic (charge) and cathodic (discharge) waves of dQ/dV curves are elated, respectively, to the Li-extraction/insertion concomitant with Ni^2+^ ⇒ Ni^4+^ redox and reversible phase transitions [17,48,49], are preserved during cycling for these Mo-doped electrodes implying, thus, their faster kinetics. For undoped NCM85, the kinetics is sluggish especially in the discharge process such as peak potentials during cycling continuously shift by 60 mV to lower values for the monoclinic-to-hexagonal M to M + H2 and by 130 mV for hexagonal-to-hexagonal H2 to H2 + H3 phase transitions, respectively. In addition, the kinetics is sluggish for these undoped electrodes in the charge process exhibiting featureless dQ/dV profiles with lower peaks intensities and much lower reversibility of the coexisted H2 + H3 phases to H3 transition responsible for the structural stability of Ni-rich NCM materials upon prolonged cycling [17].

We confirm the improved kinetics of NCM85 Mo-doped electrodes from comparative analysis of impedance spectra in Figure 10a (full scale) and b (enlarged view). It represents typical Nyquist plots measured after 15 cycles upon charging (at OCV = 4.0 V; 30 °C) from NCM85 undoped and doped samples (as indicated) in three-electrode cells. They comprised NCM85 working, Li-counter and Li-reference electrodes, thus, the impedance relates solely to the positive electrode. Though spectra in Figure 10 are typical for composite Li-intercalation cathodes [22,50] interpretation of various time constants related to different processes of these electrodes is complicated and it is still far from being unique even after many years of research in the field [51,52].

Note that the spectrum of the undoped NCM85 consists of an arch and a depressed semicircle at high frequencies (from 100 kHz to several hundred Hz), a semicircle at intermediate frequencies, and a tail at low frequencies (up to 5 mHz) ascribed to the resistance of the Li-ions solid-state diffusion in the bulk [53,54]. The spectrum measured from the Mo-doped electrode in Figure 10 is similar to that of the undoped one, however, its total impedance is a few-times lower. This is in agreement with our previous findings [5,22] demonstrating lower impedance of doped NCM materials in Li-cells. We propose the following tentative interpretation of the impedance spectra along with an equivalent circuit model of these spectra shown in Figure 10. The high-frequencies features can be ascribed to the contact resistance R_cont_ of the composite electrode, overlapped with the resistance of the Li-ions migration through the surface layer (R_sl_) or so called cathode–electrolyte interface formed during cycling [50,55,56,57], and the resistance related to the Li-ion transport across grain boundaries (R_GB_), respectively. A well-developed semicircle at intermediate frequencies in spectra of undoped and Mo-doped electrodes can be attributed to the interfacial (electrode/solution) charge-transfer resistance (R_ct_) estimated to be 6.60 and 0.76 Ohm, respectively. A tail in the impedance spectra at low frequencies (measured up to 5 mHz) is related to the Li-ions solid-state (bulk) diffusion, or Warburg impedance (W). The corresponding capacitances or constant-phase elements of these RC-constants were calculated as 1.2 and 2.6 mF/cm^2^ for undoped and doped NCM85 samples. These values are similar to those obtained for NCM85 electrodes in our recent work [54]. We also estimated the resistances related to the grain boundaries in undoped and Mo-doped electrodes as 0.30 and 0.11 Ohm, respectively. Our results clearly show that resistances R_ct_ and R_GB_ are a few-times lower for the Mo-doped electrodes. Consequently, the much lesser R_ct_ and, as a result higher the exchange current i_0_ ~ 1/R_ct_ for these electrodes, can be explained by facilitating the Li^+^ ions and electrons transport at the interface (as expected for the Mo-doped NCM materials). This is due to the additional conduction band formed near the Fermi level because of the significantly increased density of states of Ni^2+^ in the Mo-doped NCM samples [6,27]. The conductivity of grain boundaries that is inversely proportional to R_GB_ [58], is higher in this Mo-doped NCM85 material compared to that of the undoped implying, thus, improved transport characteristics and electrochemical performance—higher rate capability and lower voltage hysteresis. Furthermore, the presence of highly charged dopant Mo^6+^ segregated at the GBs (as shown in Figure 3b,c) can enhance the grain boundary conductivity similarly to that established for instance, in the lithium lanthanum titanates doped with Nb^5+^ cations [58]. Thus, Mo-doped NCM samples are appropriate examples demonstrating that the surface properties, tuned electronic structure (due to the additional conduction band near the Fermi level) and grain boundary structure of Ni-rich cathodes can be tailored by cationic doping. Relating to the preferential segregation of the Mo-dopant at the GBs, it may probably result in the formation of chemically-ordered structures across GBs [42] that strongly influence the ions and electrons transport, thus, leading to improved electrochemical performance of NCM85 in Li-cells. Another important approach for stabilization of cycling behavior and structural transformations of Ni-rich electrodes is infusion of solid electrolytes into the grain boundaries, for instance Li_3_PO_4_ into LiNi_0.76_Mn_0.14_Co_0.10_O_2_ [59]. The authors of a recent study on the Mo-doped LiNi_0.815_Co_0.15_Al_0.035_O_2_ materials [32] also explain their enhanced rate capability and faster kinetics by the infusion of the Mo-containing species Li-Mo-O into the grain boundaries and by the surface-to-bulk gradient of the Mo-dopant concentration. The last finding is in line with the Mo-segregation phenomena established in the present work (Figure 3) as well as in our previous investigations [5,27]. The above methods—cationic doping and grain boundaries modification—as well as other alternative strategies for improving cycling stability, and structural integrity of Ni-rich cathodes (surface modifications with coatings or simultaneous doping and coating and engineering of core-shell and gradient materials) though well developed, still require further studies in the fields of materials chemistry, electrochemistry, and energy storage [60].

## 4. Conclusions

In this work, we successfully synthesized Ni-rich cathode materials LiNi_0.__85_Co_0.__05_Mn_0.__0.5_O_2_ undoped and Mo-doped using the solution-combustion reaction followed by annealing the product at 760 °C under pure oxygen. Based on our previous systematic work on NCM523 and NCM811 materials, the doping level with high-charged cation Mo^6+^ was also set up by 1 at. % substituting Ni in NCM85. From structural analysis by XRD and computational considerations, we concluded that Mo preferably resides at Ni-sites. The main structural results of a minor molybdenum doping are as follows: slightly increased the lattice *a-* and *c*-parameters (likely due to the incorporation of Mo^6+^), the same value of the *c/a* ratio, and the degree of the cationic mixing. Additionally, clear splitting of the (108)/(110) X-ray diffraction peaks which remains unaffected and the lower value of (I_012_ + I_006_)/I_101_ ratio imply, thus, that the highly ordered layered structure of NCM85 was not only preserved but even improved due to the Mo^6+^ lattice doping. It was important to demonstrate that even by using a relatively simple synthetic mode like SCR and increasing the concentration of Ni in the cathode material to 85%, a clear and convincing stabilization effect was achieved by doping the NCM cathode material with a small amount of Mo. One of the important findings is that in LiNi_0.84_Mo_0.01_Co_0.10_Mn_0.05_O_2_ the Mo-dopant tends to segregate at the outermost surface layer of ≈15–20 Å and at the grain boundaries, as clearly demonstrated by results of our ToF-SIMS and HAADF-STEM-EDS studies. Mo-doped samples exhibited an improved electrochemical behavior, superior cycling stability (≈15% higher capacity retention), much lower (by ≈60%) voltage hysteresis and its evolution, higher rate capability, and faster kinetics reflected by more reversible dQ/dV profiles and lower impedance measured in Li-cells with EC-EMC/LiPF_6_ solutions. We correlated these findings with the modified electrode/solution interface due to the Mo-segregation at the surface and enrichment of the grain boundaries with the dopant. Doped LiNi_0.84_Mo_0.01_Co_0.10_Mn_0.05_O_2_ exhibited lower reactivity with the battery solutions in terms of less (by ≈25%) heat evolved, as confirmed by DSC studies at 25–350 °C, an important common characteristic of Mo-doped NCM samples established for the Ni-rich materials.

## Figures and Tables

**Figure 1 materials-14-02070-f001:**
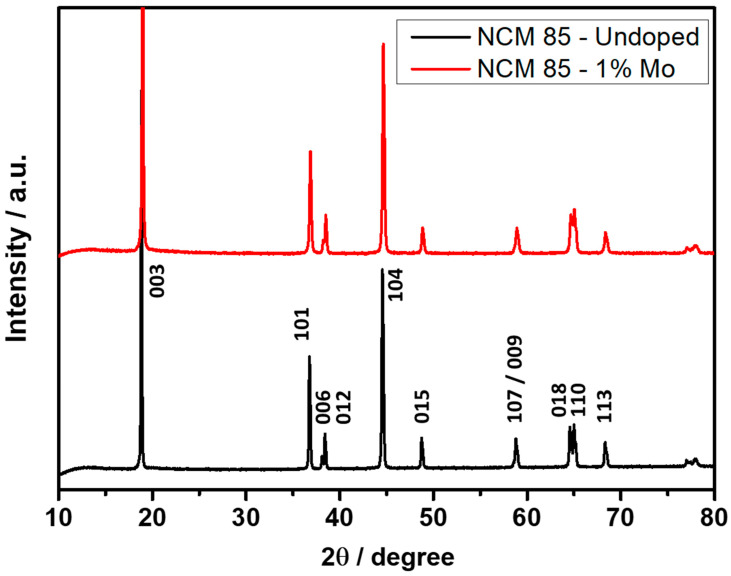
XRD patterns of NCM85 undoped and Mo-doped materials synthesized by solution-combustion method and annealed in a tubular furnace at 760 °C in pure oxygen for 6 h.

**Figure 2 materials-14-02070-f002:**
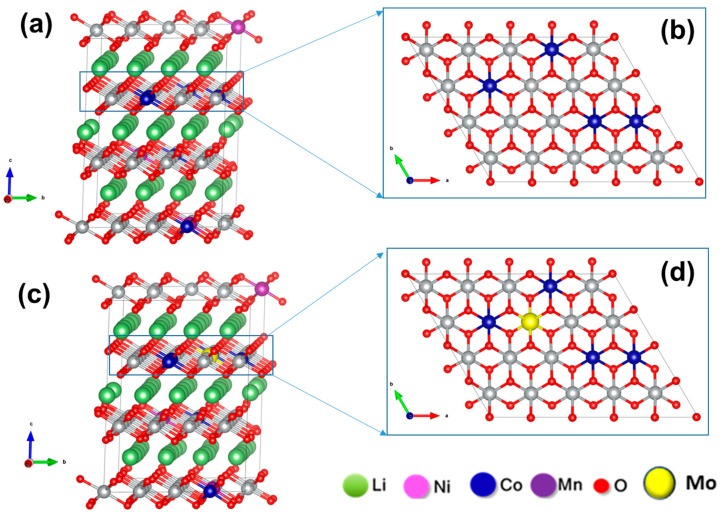
Schematic diagram for a probable configuration of undoped (**a**,**b**) and Mo-doped (**c**,**d**) NCM85 materials. (**a**,**c**) are shown in *bc-*plane, and (**b**,**d**) are shown in *ab-*plane.

**Figure 3 materials-14-02070-f003:**
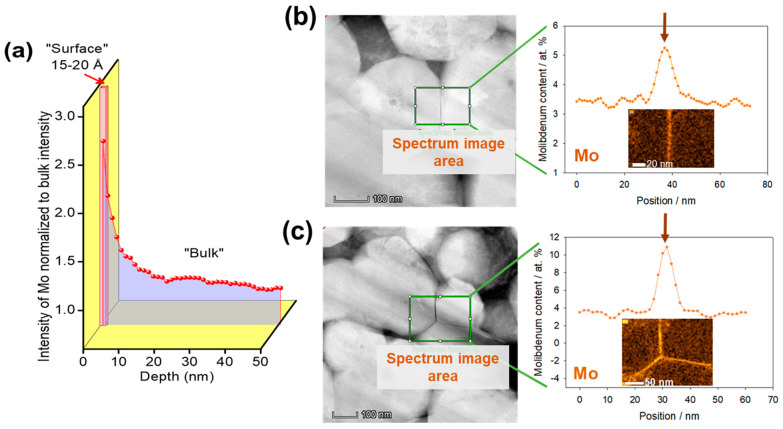
(**a**) The depth profiles of Molybdenum obtained from ToF-SIMS studies of NCM85 Mo-doped material demonstrating the dopant segregation at the outermost surface layer of ~15–20 Å. Note that due to high roughness of the powder NCM85 samples, “surface” and “bulk” regions are shown only schematically, as an eye guide. (**b**,**c**) HAADF-STEM images and the corresponding EDX maps of the Mo-K*_α_* measured from NCM85 Mo-doped material demonstrating the enrichment of the grain boundaries with the dopant. Spectral image areas are shown with green-line rectangles. The grain boundaries are marked with arrows.

**Figure 4 materials-14-02070-f004:**
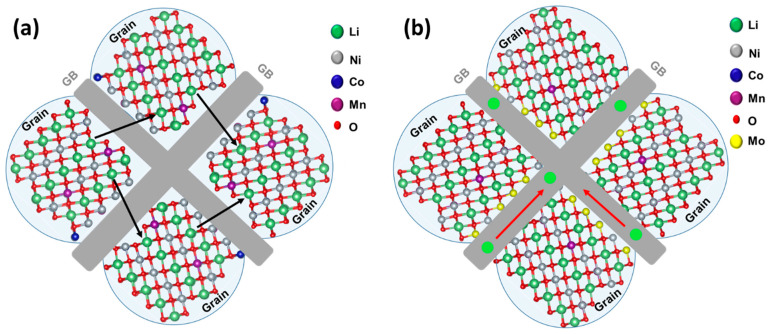
Schematic representation of possible (**a**) granular Li-ion conduction pathways in NCM85 and (**b**) Li-ion conduction pathways along grain boundaries (GBs) in Mo-doped NCM85. In these samples, the dopant segregates to the surface and the GBs are enriched with the dopant as demonstrated in Figure 3.

**Figure 5 materials-14-02070-f005:**
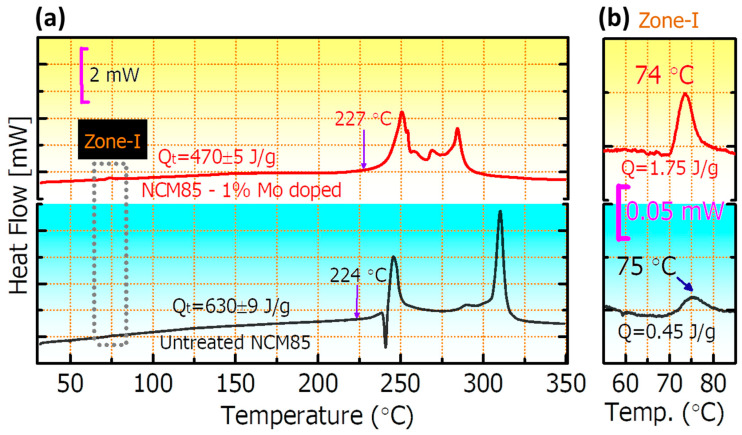
(**a**) DSC profiles measured in reactions with EC-EMC/LiPF_6_ battery solutions (LP57) of NCM85 undoped and molybdenum-doped materials, as indicated. The total heat evolved Q_t_ and the onset temperatures (indicated with arrows) are also shown; (**b**) enlarged temperature range of 55–85 °C, demonstrating the characteristic exothermal peak at ~75 °C assigned to the possible reactions of surface species like Li_2_CO_3_ and Li_2_MoO_4_ with solution. The main thermo-chemical reactions between NCM85 and solution species take place at 220–320 °C.

**Figure 6 materials-14-02070-f006:**
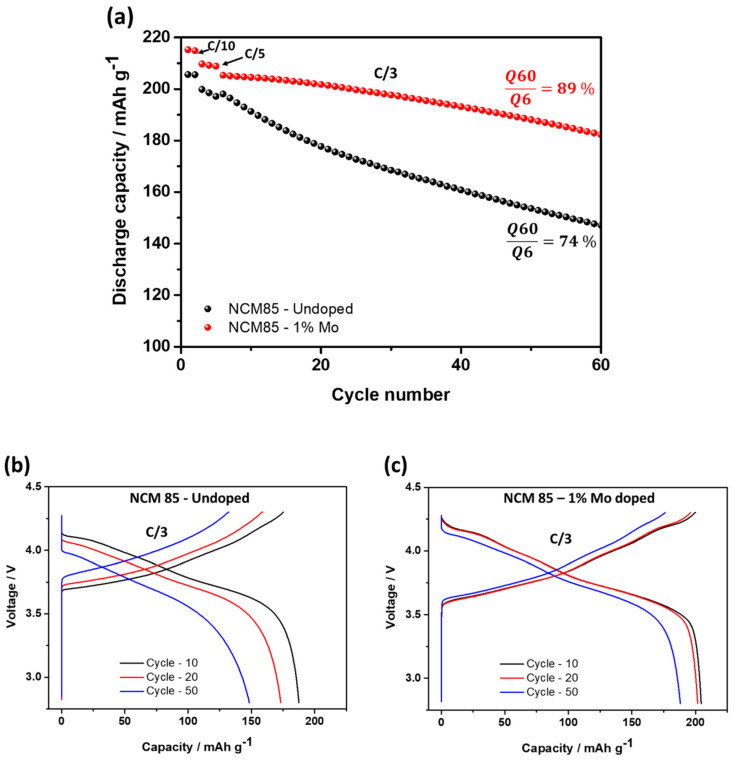
(**a**) Cycling behavior of electrodes comprising undoped and 1 at. % Mo-doped NCM85 materials, at 30 °C. Indicated are C-rates used and the capacity retention values measured for cycles 6-th and 60-th; (**b**,**c**) Voltage profiles of the above electrodes registered during charge/discharge in cycles 10-th, 20-th and 50-th at a C/3 rate. They show higher capacities obtained from Mo-doped NCM85 as well as more reversible phase transitions at ~4.2 V of these electrodes compared to the undoped ones.

**Figure 7 materials-14-02070-f007:**
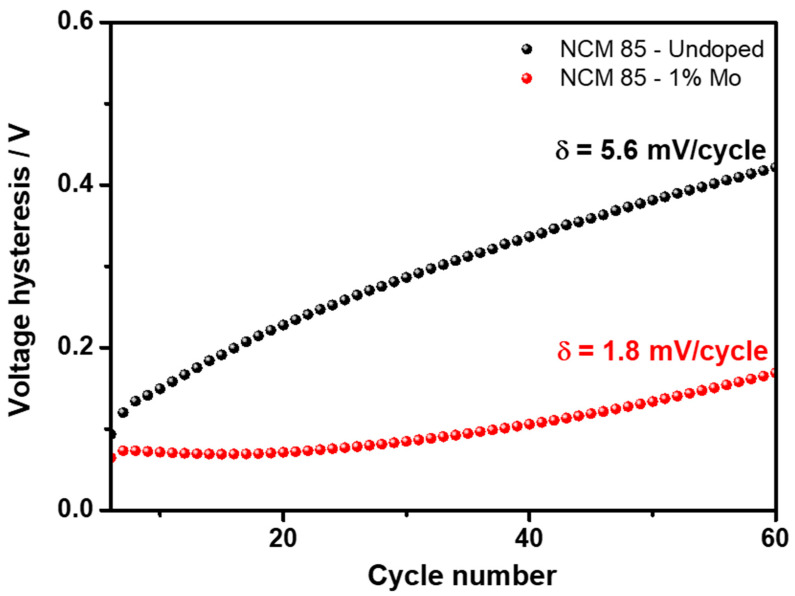
The voltage hysteresis measured from cycling performance (Figure 6a) of electrodes comprising undoped and 1 at. % Mo-doped NCM85 materials, at 30 °C. Evolution of the voltage hysteresis (δ, mV/cycle) is indicated.

**Figure 8 materials-14-02070-f008:**
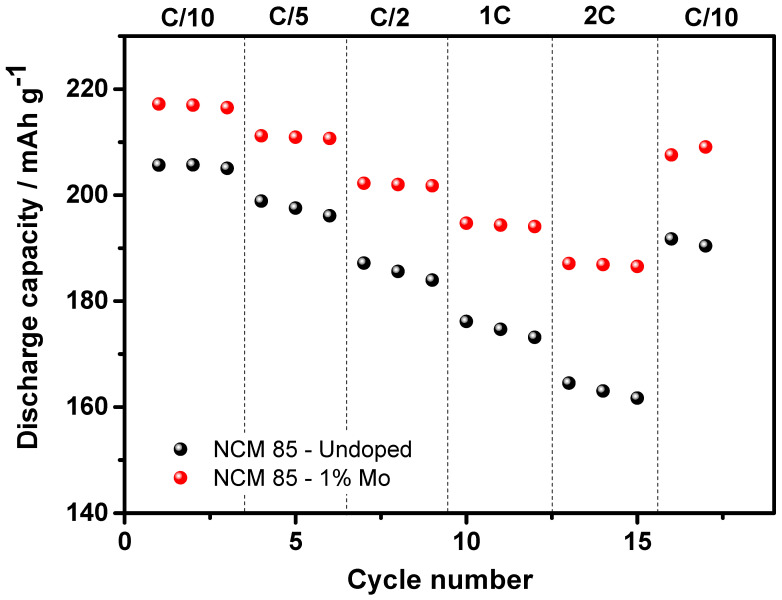
Discharge capacities measured at various C-rates (C/10, C/5, C/2, 1C, and 2C) from electrodes comprising undoped and 1 at. % Mo-doped NCM85 materials, at 30 °C.

**Figure 9 materials-14-02070-f009:**
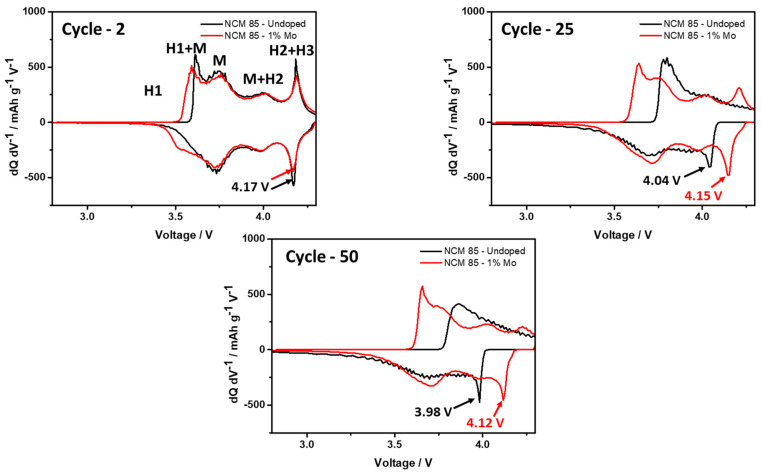
Differential capacity dQ/dV as a function of the cell voltage measured from electrodes comprising undoped and 1 at. % Mo-doped NCM85 materials, at 30 °C for cycles 2, 25, and 50.

**Figure 10 materials-14-02070-f010:**
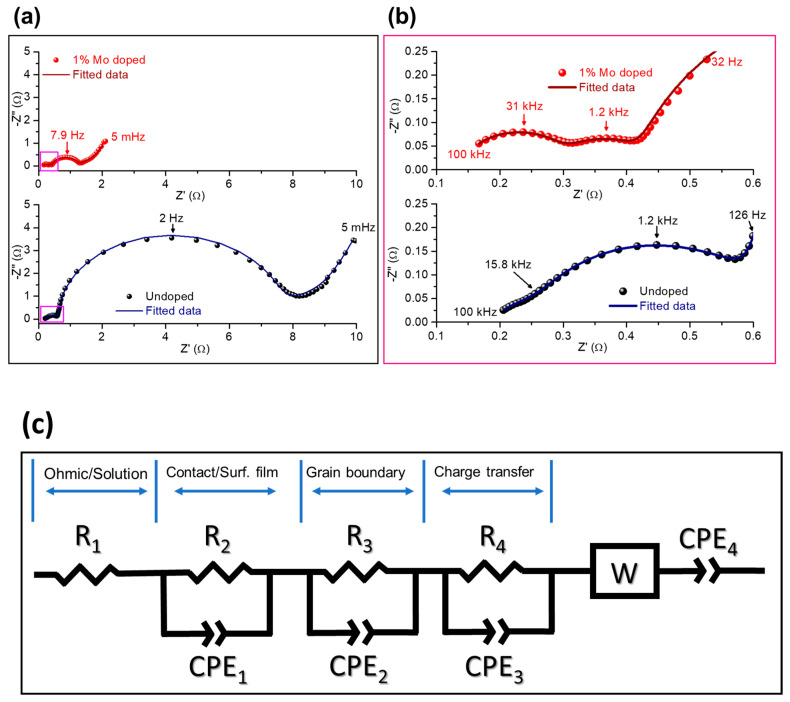
Impedance spectra of NCM85 undoped and Mo-doped electrodes measured during charging at 4.0 V, 30 °C in three-electrode pouch-type cells with the Li-reference electrodes (as described in the text). Symbols are experimental points and solid lines are fitted data: (**a**) full scale spectra and (**b**) an enlarged view of the high-to-medium frequency regions of the spectra marked with the rose color rectangles. (**c**) An equivalent circuit model of the impedance of NCM85 electrodes.

**Table 1 materials-14-02070-t001:** Chemical composition (in at. %) of NCM85 undoped and Mo-doped materials measured after their synthesis and annealing at 760 °C under pure oxygen for 6 h.

Sample	Ni	Co	Mn	Mo
**NCM85—Undoped**	85.4	9.8	4.8	-
**NCM85—1 at. % Mo doped**	84.1	10.0	4.9	0.96

**Table 2 materials-14-02070-t002:** The lattice parameters *a* and *c*, mean coherent domain size, cation mixing (Ni^2+^/Li^+^) and intensity ratios of 003, 104, and 012, and the 006 and 101 planes obtained from XRD patterns of NCM85 undoped and 1 at. % Mo-doped materials**.**

Samples	*a*/Å	*c*/Å	*c*/*a*	Mean Coherent Domain Size/nm	Ni^2+^ in Li-Layer	I_003_/I_104_	(I_012_ + I_006_)/I_101_
**NCM85—Undoped**	2.87002(3)	14.1882(3)	4.94	123.2(9)	0.050(5)	1.149	0.453
**NCM85—1 at. % Mo doped**	2.87343(3)	14.1920(3)	4.94	81.1(9)	0.050(5)	1.257	0.419

**Table 3 materials-14-02070-t003:** Anodic and cathodic peaks potentials (V vs. Li/Li^+^) related to the Ni^2+^/Ni^4+^ redox and M+H2 and H2+H3 phase transitions measured for undoped and 1 at. % Mo-doped electrodes upon cycling at C/10 (cycle 2) and C/3 rates (cycles 5, 10, and 25).

	NCM85-Undoped		NCM85-1% Mo Doped	
**Charge**	**M + H2**	**H2 + H3**	**M + H2**	**H2 + H3**
Cycle 2	4.00	4.18	4.01	4.19
Cycle 5	4.00	4.20	4.02	4.19
Cycle 10	4.02	4.21	4.02	4.21
Cycle 25	4.03	4.23	4.02	4.21
**Discharge**	**M + H2**	**H2 + H3**	**M + H2**	**H2 + H3**
Cycle 2	3.98	4.17	3.99	4.17
Cycle 5	3.98	4.15	3.98	4.16
Cycle 10	3.96	4.09	3.98	4.16
Cycle 25	3.92	4.04	3.98	4.15

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
