# Peer review of "Studies of Nickel-Rich LiNi0.85Co0.10Mn0.05O2 Cathode Materials Doped with Molybdenum Ions for Lithium-Ion Batteries"

_materials, 2021, doi:10.3390/ma14082070_

Round 1
Reviewer 1 Report
The authors report “Studies of Nickel-Rich LiNi0.85Co0.10Mn0.05O2 Cathode Materials Doped with Molybdenum Ions for Lithium-Ion Batteries”. Although the work presented is interesting to materials research community, this manuscript needs to be modified, please find the major concerns below.
- The authors did not utilize XRD data completely to explain some critical characteristics of the materials synthesized. It seems there is a slight peak position shifted towards higher 2 theta as it supposed to be since Mo(6+) is smaller ion radii when compared to Ni(2+) but almost similar to Ni(3+). This could tell some story here to provide proof of alloying/doping effect by Mo into the existing crystal. For example, please see DOI: 10.1039/c8nr04399j.
- The authors mentioned in the manuscript that Ni is replaced by Mo but at what oxidation states are those? Please mention this and make it clear to the readers.
- Also, I suggest the authors should estimate the crystallite sizes using Scherrer’s formula using XRD diffraction peaks and this provide detail of average crystallite sizes of the materials synthesized. For example, please see https://doi.org/10.1021/la0477183 and cite this relevant literature.
- Did the authors try to incorporate more Mo into the synthesized cathode material? Mo can bring better synergistic effects to the catalyst as we know from the existing catalysis literature.
Author Response
Comment 1. The authors did not utilize XRD data completely to explain some critical characteristics of the materials synthesized. It seems there is a slight peak position shifted towards higher 2 theta as it supposed to be since Mo(6+) is smaller ion radii when compared to Ni(2+) but almost similar to Ni(3+). This could tell some story here to provide proof of alloying/doping effect by Mo into the existing crystal. For example, please see DOI: 10.1039/c8nr04399j.
Comment 2. The authors mentioned in the manuscript that Ni is replaced by Mo but at what oxidation states are those? Please mention this and make it clear to the readers.
Answer to Comments 1 and 2. The results presented in Table 2 of our paper show that the unit cell parameters for Mo-substituted (doped) sample are higher than for the undoped one despite the smaller ionic radius of Mo6+. The increase of unit cell parameters is due to the slight compensative increase of the amount of Ni2+ in doped sample in order to keep the electro-neutrality of the material. This fact is an indication for Mo incorporation in the structure. Having in mind the synthesis conditions (760 oC in pure oxygen during 6 hours), the oxidation state of molybdenum is assumed to be 6+. In addition, we have shown in our previous combined experimental and computational modelling studies [Susai et. al., ACS Appl. Energy Mater. 2019, 2, 6, 4521–4534; Bruer et. al. ACS Appl. Mater. Interface 2018 ] that Mo ions replaced Ni ions out of other transition metal ions in Mo-doped NCM811 and NCM523 with the same dopant concentration of 1 at. %. From DFT calculations, we concluded Mo-ion was in 6+-oxidation state and these highly charged Mo-ions changed the distribution of Ni-ions having different oxidation states and effectively Ni3+ ions were reduced upon Mo-doping. Therefore, we believe the same would be true in the case of 1% Mo-doped NCM851005 studied in the current work. We incorporated this explanation in the revised version (p. 7).
Comment 3. Also, I suggest the authors should estimate the crystallite sizes using Scherrer’s formula using XRD diffraction peaks and this provide detail of average crystallite sizes of the materials synthesized. For example, please see https://doi.org/10.1021/la0477183 and cite this relevant literature.
Answer 3. We have calculated the mean coherent domain size (crystallite size) of NCM85 undoped and Mo-doped samples and represented them in Table 2. The crystallite size was obtained by the analysis of the diffraction lines broadening of all lines presented in the diffraction pattern. For this purpose, the profiles of the diffraction peaks were fitted by means of fundamental parameters approach implemented in the Topas-4, where the diffraction line is presented as convolution of the instrumental and the specimen functions. The instrumental function itself is also a convolution of the line broadening functions of the X-Ray source and the slit elements stacked on the optical path of the X-Ray beam. This function is strictly individual for the optical configuration of the diffractometer used to collect the diffraction pattern data. The method used in the present paper is more advanced than the proposed Scherrer’s method used by Holger Borchert et al. in their paper https://doi.org/10.1021/la0477183 Langmuir 2005, 21, 1931-1936, “Determination of Nanocrystal Sizes: A Comparison of TEM, SAXS, and XRD Studies of Highly Monodisperse CoPt3 Particles”. That is why, we consider not to cite this ref. in our paper. Instead, we included the following reference [40]: Bruker AXS (2008): TOPAS V4: General profile and structure analysis software for powder diffraction data. - User's Manual, Bruker AXS, Karlsruhe. http://algol.fis.uc.pt/jap/TOPAS%204-2%20Users%20Manual.pdf
Comment 4. Did the authors try to incorporate more Mo into the synthesized cathode material? Mo can bring better synergistic effects to the catalyst as we know from the existing catalysis literature.
Answer 4. We have not tried to incorporate more Mo into NCM85 since it was important to preserve the high Ni content close to 85 at. % in the material. In our recent study of NCM811 cathode material [ref. 5,35], we have also incorporated 1 at. % Mo in LiNi0.79Mo0.01Co0.1Mn0.1O2 as well as increased its content to 3 at. % Mo in LiNi0.77Mo0.03Co0.1Mn0.1O2 electrodes. In addition, 2 at. % Mo-doped and 3 at. % Mo-doped (by Ni and Mn partial substitution) materials were studied:
LiNi0.79Mo0.01Co0.1Mn0.09Mo0.01O2 and LiNi0.78Mo0.02Co0.1Mn0.09Mo0.01O2.
We have established that doping of NCM85 material with minor Mo concentration of around 1.0 at. % resulted in the enhancement of their electrochemical cycling performance, lowering impedance, heat evolution in reactions with battery solutions, and stabilization of reversible structural transformations. This explanation is incorporated into the revised version (p. 3).

Reviewer 2 Report
The manuscript, “Studies of Nickel-Rich LiNi0.85Co0.10Mn0.05O2 Cathode Materials Doped with Molybdenum Ions for Lithium-Ion Batteries” reports the effects of doping a nickel-rich NMC85 cathode with a small concentration of molybdenum and compares the electrochemical results between doped and undoped NMC85 samples. The results are interesting, and the characterization reported in this manuscript is sufficient to support the conclusions. However, there are some missing details that need to be provided and a few very minor concerns in the manuscript that need to be addressed before it can be published in Materials. They are listed as follows:
- This manuscript does not explain how and why the concentration of 1% was chosen. The authors have carried out previous studies that explains the rationale behind this choice (Ref 5). It would be good if the authors could explicitly describe this rationale (in brief) in the Introduction section of this manuscript so that this manuscript could serve as a standalone article.
- Could the authors include a comparison of the voltage profiles of the undoped and doped sample at initial and later cycles (say 30th and 50th)? This would provide a clearer picture of where exactly the capacity fading differs in both samples.
- In the figure caption of Figure 7: “The voltage hysteresis measured from cycling profiles (Figure 5)…”, I believe the correct figure number within the parentheses should be “Figure 6”.
Author Response
Comment 1. This manuscript does not explain how and why the concentration of 1% was chosen. The authors have carried out previous studies that explains the rationale behind this choice (Ref 5). It would be good if the authors could explicitly describe this rationale (in brief) in the Introduction section of this manuscript so that this manuscript could serve as a standalone article.
Answer 1. We included the following sentence in the Introduction section (p. 3): From preliminary studies, we observed that Mo-doping level as small as 1 at. % can be considered as optimal in NCM85 material to enhance the electrochemical cycling performance, to lower electrode impedance, heat evolution in reactions with battery solutions, and to stabilize reversible structural transformations.
Comment 2. Could the authors include a comparison of the voltage profiles of the undoped and doped sample at initial and later cycles (say 30th and 50th)? This would provide a clearer picture of where exactly the capacity fading differs in both samples.
Answer 2. We have included Figure 6 b and c in the revised paper (p. 12) that compares the voltage profiles registered during charge/discharge of undoped and Mo-doped samples at cycles 10th, 20th and 50th, at a C/3 rate.
Comment 3. In the figure caption of Figure 7: “The voltage hysteresis measured from cycling profiles (Figure 5)…” I believe the correct figure number within the parentheses should be “Figure 6”.
Answer 3. Thank you. The correct figure number in the parentheses should be “Figure 6”. We have corrected this in the Revised version.
